# Ultrabright and stable top-emitting quantum-dot light-emitting diodes with negligible angular color shift

Mengqi Li[1], Rui Li[1], Longjia Wu[2], Xiongfeng Lin[2], Xueqing Xia[1], Zitong Ao[2], Xiaojuan Sun[1], Xingtong Chen[1] & Song Chen[1,3] ✉

Top emission can enhance luminance, color purity, and panel-manufacturing compatibility for emissive displays. Still, top-emitting quantum-dot light-emitting diodes (QLEDs) suffer from poor stability, low light outcoupling, and non-negligible viewing-angle dependence because, for QLEDs with non-red emission, the electrically optimum device structure is incompatible with single-mode optical microcavity. Here, we demonstrate that by improving the way of determining reflection penetration depths and creating refractive-index-lowering processes, the issues faced by green QLEDs can be overcome. This leads to advanced device performance, including a luminance exceeding 1.6 million nits, a current efficiency of 204.2 cd A$^{-1}$, and a $T_{95}$ operational lifetime of 15,600 hours at 1000 nits. Meanwhile, our design does not compromise light outcoupling as it offers an external quantum efficiency of 29.2% without implementing light extraction methods. Lastly, an angular color shift of $\Delta u'v' = 0.0052$ from 0° to 60° is achieved by narrowing the emission linewidth of quantum dots.

Colloidal quantum dots are high-performance, cost-effective light sources for the next generation of display technologies[1]. Quantum-dot light-emitting diodes (QLED) that emit light from the bottom substrate have achieved high external quantum efficiency (EQE)[2–4]. However, to meet the requirements of traditional and emerging applications, QLEDs need even higher luminance and longer operational lifetime[5], which are more directly determined by current efficiency (CE) than EQE. While the multijunction (or tandem) structure is a standard CE-enhancing solution, adding functional layers and interfaces significantly reduces device stability[6,7]. The display industry favors the top emission structure for CE enhancement because of its compatibility with panel-manufacturing processes and high aperture ratio[8]. However, previously demonstrated devices have struggled to achieve high optical and electrical performance simultaneously. Optically, balancing the microcavity effect and total light outcoupling to avoid the

tradeoffs between CE, EQE, and angular dependence is a primary criterion for top emission[9]. The integrated emission rate enhancement due to the microcavity effect ($G_{int}$) is formulated using Eq. 1[10],

$$G_{\mathrm{int}} = \frac{\xi}{2}\frac{2}{\pi}\frac{1-R_1}{1-\sqrt{R_1R_2}}\sqrt{\pi\ln 2}\frac{\lambda}{\Delta\lambda_{\mathrm{n}}}\frac{\lambda_{\mathrm{cav}}}{L_{\mathrm{cav}}}\frac{\tau_{\mathrm{cav}}}{\tau} \qquad (1)$$

Here, $\xi$ is the antinode enhancement factor. $R_1$ and $R_2$ are the reflectivity of the two mirrors ($R_1 < R_2$), $\lambda$ is the emission wavelength, $L_{\mathrm{cav}}$ is the cavity length, $\tau_{\mathrm{cav}}$ and $\tau$ are QDs' spontaneous emission lifetimes inside and without a microcavity, respectively. The equation suggests the shortest possible $L_{\mathrm{cav}}$ should be prioritized because a single optical mode can have the largest integral overlap with QDs' emission spectrum. However, previous attempts have failed to achieve this goal except for a red device[11]. Multi-mode cavities, e.g., a $L_{\mathrm{cav}} = 3/2\lambda$ design,

[1]Suzhou Key Laboratory of Novel Semiconductor-optoelectronics Materials and Devices, College of Chemistry, Chemical Engineering and Materials Science, Soochow University, Suzhou 215123 Jiangsu, China. [2]TCL Corporate Research, 1001 Zhongshan Park Road, Nanshan District, Shenzhen 518067 Guangdong, China. [3]Jiangsu Key Laboratory of Advanced Negative Carbon Technologies, Soochow University, Suzhou 215123 Jiangsu, China. ✉e-mail: songchen@suda.edu.cn

reduce the optical enhancement multiple times[12,13]. Moreover, there is a common concern from the field of organic light-emitting diodes (OLEDs) that single-mode cavity causes substantial angular-dependent color shifts[14]. However, simulations predicted that QLEDs could potentially achieve negligible color shifts thanks to much narrower emission linewidths ($\Delta\lambda_n$) than organic emitters[15].

Besides the optical penalty, most top-emitting (TE) designs adopted for QLEDs also compromise their electrical performance. The low-mobility semiconducting layers in the best-performing QLEDs are generally over 100 nm thick[16,17]. This creates optical distances that are too large for single-mode microcavities with green or blue emitters, which have relatively short emission wavelengths. Deviating from the optimal structure to meet the resonance condition compromises charge balance and complicates the degradation mechanism[18]. As a result, except for red emission, TE QLEDs have not yet demonstrated operational lifetimes that are competitive with those of their bottom-emitting (BE) counterparts and TE OLEDs[19–22].

In this study, we use TE green QLED, the primary contributor of white field luminance, as an example to demonstrate how to achieve high emission rate enhancement, stable electrical performance, high light outcoupling rate, and negligible angular color shift simultaneously. The correct determination of reflection penetration depths has resolved the theoretical confusion surrounding the creation of single-mode green QLEDs. Accurately modulating the refractive indices of nanocrystal-assembled films enables us to combine the most electrically advanced design with the single-mode cavity. To address the common concerns of the single-mode design, we have enhanced the total light

outcoupling by limiting the top mirror's reflectivity and suppressed the angular color shift to a negligible level by synthesizing green QDs with natural emission linewidth smaller than common values. As a result, the single-mode green QLED exhibits all-round leading performance.

## Results

### Design of single-mode green QLED

Figure 1a, b show the schematic illustration and transmission electron microscopy (TEM) image of ZnCdSeS green QDs, respectively. A 6 nm ZnCdSe core is surrounded by a 1.5 nm intermediate shell of ZnCdS, over which an outer shell of ZnS (1 nm) is grown, enabling reduced auger recombination and enhanced carrier injection[5,16]. Oleic acid ligands bind to the QD surface for defect passivation and colloidal stability. These QDs exhibit an improved photoluminescence quantum efficiency (PLQE) of 86.2% in the solution and an internal PLQE of 90.3% in solid films (see additional data and the calculation method for internal PLQE in Supplementary Fig. 1).

Figure 1c shows the cross-sectional TEM image of our TE structure. The semiconducting layer stacks are nearly identical to the state-of-the-art BE devices, except that a 10 nm thick ITO layer was deposited to protect the bottom Ag electrode from PEDOT:PSS and improve wetting. Unlike most previous work, we ruled out dielectric materials as top mirrors because the considerable field penetration depth is incompatible with the single-mode design. Thin metal films offer less field penetration and moderate reflectivity. However, the discussion on the penetration depth into metallic mirrors has been confusing due to inconsistent sign conventions[23].

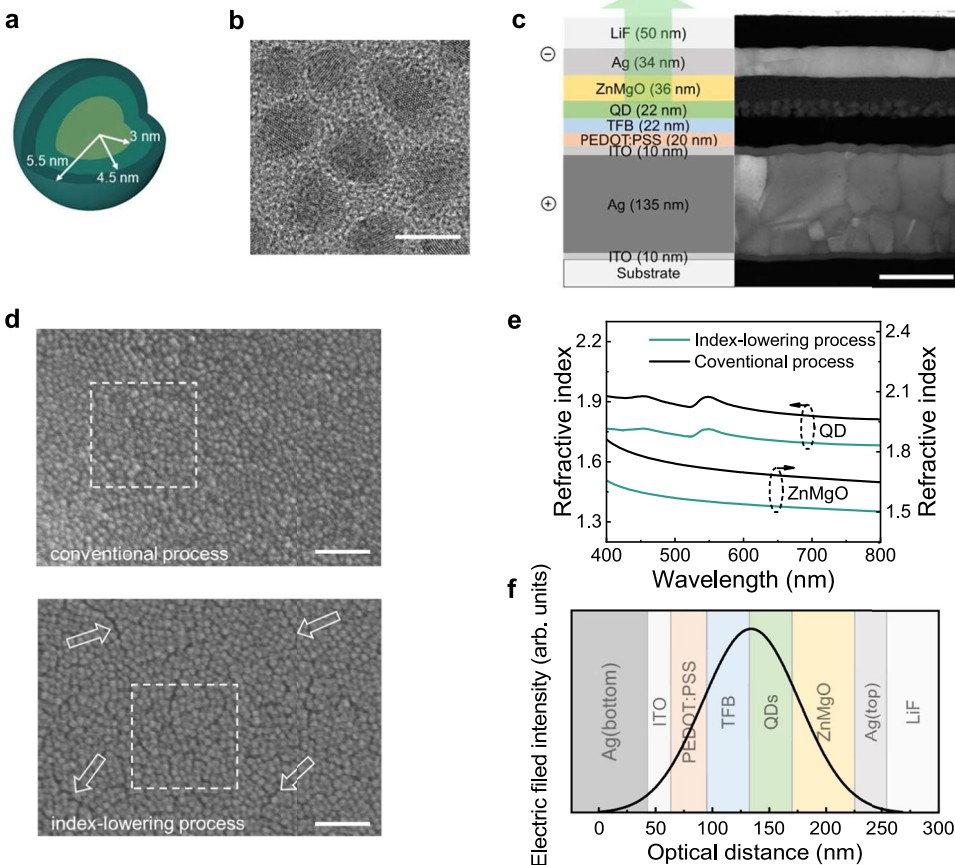

**Fig. 1 | Quantum dots and device structure. a** Illustration of the QD structure. **b** TEM image of QDs (scale bar = 10 nm). **c** Illustration and cross-sectional TEM image of the top-emitting QLED structure (scale bar = 100 nm). **d** SEM images of QD layers' surface morphology. The conventional and index-lowering processes result in nanocrystal areal densities of 53.5 ± 1.6 and 47.8 ± 2.0 per 10,000 nm² within the

dashed box, respectively. The arrows point to some of the gaps between QDs caused by the index-lowering processes (scale bar = 100 nm). **e** Refractive indices of nanocrystal-assembled films formed using the conventional method and index-lowering process. **f** Schematic diagram of one-dimensional field distribution inside a $\lambda/2$ cavity.

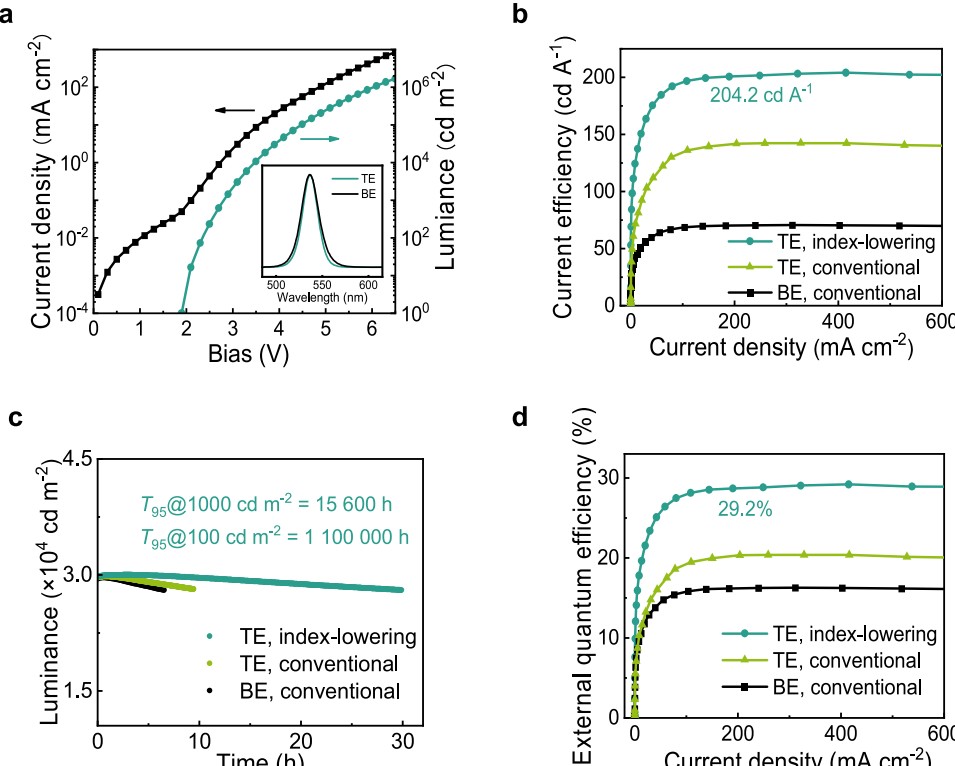

**Fig. 2 | Device performance. a** L-J-V characteristics and EL spectra. **b** Current efficiency. **c** Operational lifetime. **d** External quantum efficiency. The bottom-emitting device was fabricated using the same semiconductor stacks. The top-emitting device fabricated using the conventional process has a significantly reduced thickness to fit the single-mode resonance.

Equation 2 shows the condition of microcavity resonance at the substrate-normal direction.

$$L_{cav} = \sum_i n_i d_i + L_{pen1} + L_{pen2} = \frac{m\lambda}{2} \quad (2)$$

For every single pass, the total light path in the semiconductor media ($nd$) plus the field penetration depth into the top ($L_{pen1}$) and bottom ($L_{pen2}$) mirrors must equal $m\lambda/2$, corresponding to a $m2\pi$ phase change, where $m$ is an integer. Due to the reflective index of metals, a phase change $\pi$ should be added to the conventional expression[24], meaning the penetration depth should be expressed using Eqs. 3 and 4 rather than the ones conventionally used in the literature[25–27].

$$L_{pen} = \frac{\lambda(\pi - \varphi)}{4\pi} \quad (3)$$

$$\varphi = \pi + \arctan \frac{2n_0 k_1}{n_0^2 - n_1^2 - k_1^2} \quad (4)$$

Here, $n_0$ is the refractive index of the dielectric medium, $n_1$ and $k_1$ represent the real and imaginary parts of the metal's refractive index, respectively (see Supplementary Fig. 2 and Supplementary Table 1 for measured optical constants). In a green QLED ($\lambda = 537$ nm), the single-pass optical distance through the semiconducting stacks is over 180 nm, meaning the total penetration depth should be less than 90 nm to allow for a single-mode resonance ($m = 1$). Therefore, the conventional expression forces a long-cavity (multi-mode) design, which aggravates the optical penalty (see Supplementary Note 1 for details). In comparison, Eq. 3 produces a total $L_{pen}$ value of only 85 nm, potentially fitting the $\lambda/2$ resonance mode.

Fitting a high-performance green QLED into a single-mode microcavity remains challenging despite correctly obtaining the reflection penetration depths. This is due to the contradiction between the short cavity length ($\lambda/2 = 268.5$ nm), high refractive indices of II-VI compounds, and the minimum thickness required for electrical performance. To tackle this issue, we modified the formation environment of QD-assembled films. As seen in Fig. 1d, the conventional process produces closely packed QDs. In contrast, applying a co-solvent strategy and replacing post-thermal treatment with low-vacuum drying results in uniformly distributed gaps of several nanometers wide between QDs without disruptive cracks or pinholes. This leads to a decrease of approximately 12% in the in-plane packing density, which can be attributed to the suppression of nanocrystal reassembling[28]. As shown in Fig. 1e, the refractive indices of emissive QDs and ZnMgO nanoparticles are lowered by 0.15 and 0.17 ($\lambda = 537$ nm), respectively, which match the reduction in in-plane packing density (see the effect of mixed solvent ratio in Supplementary Fig. 2h). Consequently, the nanocrystal-assembled layers can attain sufficient thickness to ensure electrical performance. Figure 1f shows the estimated one-dimensional field distribution, in which the penetration depths into mirrors plus the optical distance through the semiconducting layers result in a value equaling $\lambda/2$.

## QLED performance

TE green QLEDs were fabricated using the index-lowering process. Figure 2a shows the L-J-V characteristics and EL spectra for the devices with the most balanced performance (see Supplementary Fig. 3 for the device fabricated with the conventional process and additional device parameters). The J-V curve and the notable sub-threshold turn-on ($V_{th} = 1.9$ V) resemble the BE QLEDs due to their nearly identical semiconducting layer stacks[29]. The microcavity's resonance wavelength was tuned to match the natural emission spectrum of QDs, with

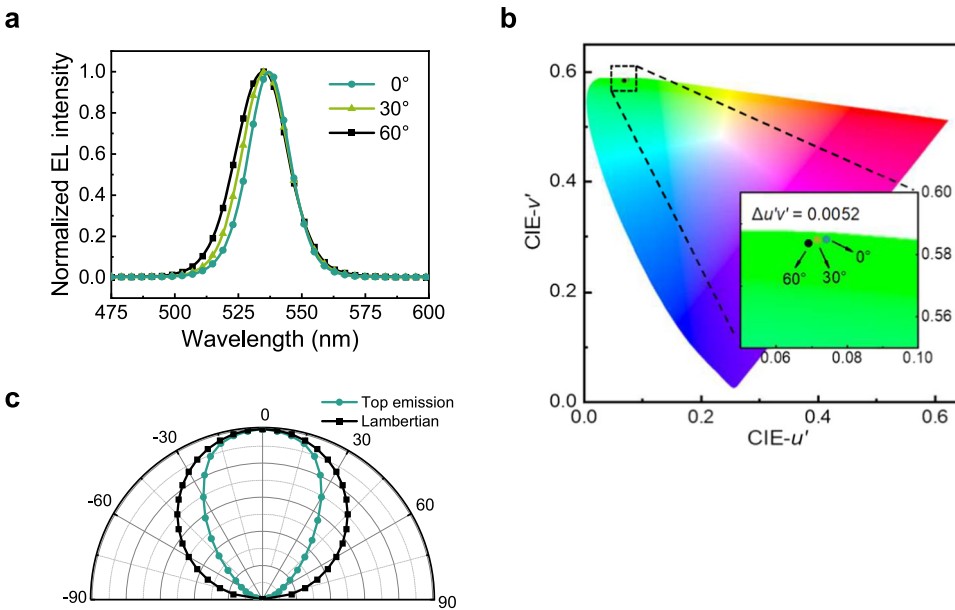

**Fig. 3 | The viewing-angle dependence. a** Emission spectra. **b** Color coordinates in the CIE 1976 uniform (u′,v′) diagram. **c** Normalized luminance intensity.

details to be discussed later. The measured spectrum centers at 537 nm with a full width at half maximum (FWHM) narrowed from 21.5 to 19.2 nm. At the substrate-normal direction, which is usually of interest, the device exhibits a luminance exceeding 1,600,000 cd m$^{-2}$ at the bias of 6.5 V. Figure 2b shows that the CE reaches a maximum value of 204.2 cd A$^{-1}$, while the conventional process offers a much lower CE. Compared with the BE device, the integrated emission enhancement due to the microcavity effect achieves 2.9. Remarkably, the optimized device exhibits the highest luminance intensity for green QLED and outperforms any previously reported QLED in current efficiency[7,21,30–33], which is important in augmented reality, virtual reality, and outdoor displays.

Our TE green QLEDs share the same semiconducting layer stacks as the BE counterparts, known for high operational stability[16]. The substantially enhanced CE allows the device to operate under a much-reduced current density and exhibit a significantly reduced degradation rate. To prevent overestimating lifetime, it was confirmed that luminance uniformity within the emissive area was maintained during all the tests. As seen in Fig. 2c, the BE device and the TE device fabricated using the conventional process show $T_{95}$ of 3400 and 5000 h at an initial luminance of 1000 cd m$^{-2}$, respectively. In comparison, the TE QLED fabricated using the index-lowering process shows a $T_{95}$ of 15,600 h at 1000 cd m$^{-2}$ and a $T_{95}$ of 1,100,000 h at 100 cd m$^{-2}$ (acceleration factor, $n = 1.85$) due to the much-reduced operating current density. Data extrapolation based on the reported method produces a $T_{50}$ of 10 million h at 100 cd m$^{-2}$ (see additional lifetime data in Supplementary Fig. 4)[34,35]. The operational lifetime of the optimized device marks a new record for green QLEDs[21].

Unlike luminance, CE, and operational lifetime, the measurement of EQE counts the photons emitted in all directions. Previously reported devices struggle to achieve high CE and EQE simultaneously. However, the shortest cavity length adopted in this work allows the top mirror for higher transmission. Figure 2d shows that our TE QLED achieves an EQE value of 29.2%, which is a 1.81-fold enhancement compared to the BE counterpart and so far the highest value for green QLEDs without a light extraction design[21,36] (see the statistical distribution of EQE in Supplementary Fig. 3). The EQE enhancement indicates that the emission rate enhancement surrounding the substrate-normal direction is more significant than the loss in larger off-axis angles, and the unprecedented CE is achieved without compromising EQE. Notably, light extraction

methods have been extensively studied in the field of OLEDs[37]. To recover optical losses and further enhance light outcoupling, microlens arrays, scattering layers, and corrugated structures could be added to this planar device in future research.

The variation of color and luminance with viewing angle is important for display. Figure 3a shows the angular-dependent emission spectra of the TE green QLED. Despite the single-mode cavity without additional light extraction design, the emission wavelength is blueshifted by only 2.5 nm from 0° to 60°. In Fig. 3b, the corresponding chromaticity coordinates are plotted in the CIE 1976 uniform diagram (u′,v′), which quantitatively evaluates color in a perceptually uniform manner. Remarkably, the single-mode QLED achieves negligible angular color shifts of Δu′v′ = 0.0030 and 0.0052 at the off-axis angles of 30° and 60°, respectively (see the presentation in the CIE 1931 non-uniform diagram (x,y) in Supplementary Fig. 5). To our best knowledge, this is the lowest color shift ever reported for TE QLEDs, even though previous work generally employed multi-mode cavities and scattering layers that are beneficial for reducing color shifts[30,36]. The color shifts of these QLEDs are also smaller than existing mobile display products and OLEDs with ultralong cavity lengths[38,39]. Figure 3c shows the angular dependence of luminance intensity. The emission profile corresponds to a Lambertian factor of 0.62π, which is even comparable with previous devices adopting a multi-mode cavity[20,40]. The near Lambertian emission (1.03π) of the BE device is shown in Supplementary Fig. 6.

## Balancing optical and electrical performance

The high-performance listed above is primarily due to achieving single-mode resonance in green QLED, as well as addressing the common concerns about single-mode design. As shown in Eq. 1, reducing QDs' spontaneous emission linewidth (Δλ) should increase its overlapping integral with the microcavity's optical mode density and enhance $G_{int}$. This effect has been largely overlooked in past reports, limiting the CE enhancement. Besides high PLQE, our green QDs feature a much narrower FWHM than previously reported ones with competitive electroluminescence performance[17]. As shown in Fig. 4a, narrowing the emission peak's FWHM from 27 to 21 nm while fixing the emission wavelength can additionally enhance CE by 15%, which is consistent with Eq. 1. In comparison, the CE of BE devices is not much affected by FWHM (see Supplementary Fig. 7). The narrow emission linewidth also

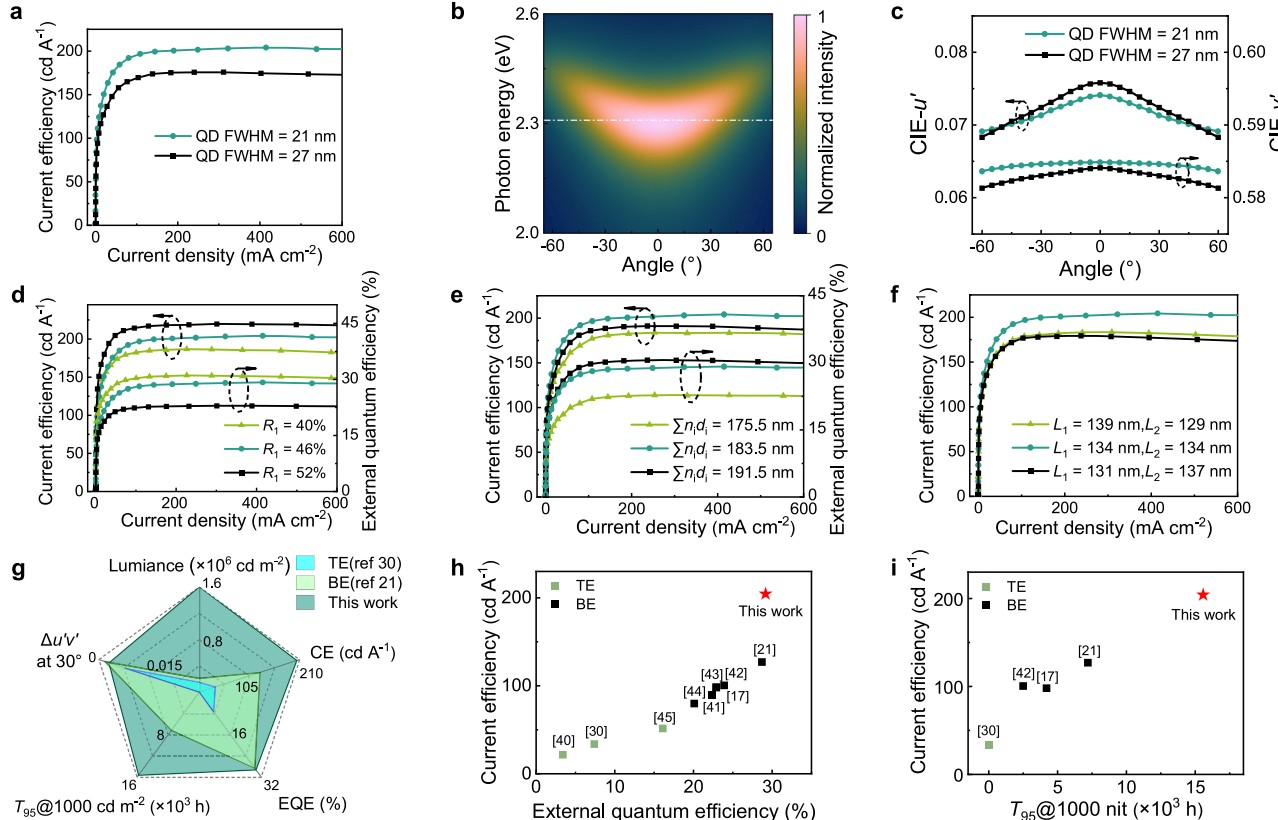

**Fig. 4 | Balancing the microcavity effect, total light outcoupling, and electrical performance. a** Effect of QDs' natural emission linewidth on current efficiency. **b** Optical mode as a function of viewing angles (white dashed line: QD emission peak). **c** Effect of QDs' natural emission linewidth on angular color shifts. **d** Effects of top mirror's reflectivity. **e** Effect of semiconducting layers' thickness. $n_i$ and $d_i$ denote the refractive index and thickness of each non-metallic layer, respectively. **f** Antinode enhancement factor effect. $L_1$ and $L_2$ denote the distance of the emitting layer to the top and bottom of the cavity, respectively (The device parameters for

*d–f* are detailed in Supplementary Table 1). **g** Radar chart summarizing the device performance. Color shift data was not provided in reference[21]. For the integrity of the bottom-emitting QLED data, we referred to the best result reported elsewhere from a similar device (see Supplementary Table 2). **h** Comparison of CE and EQE. **i** CE and lifetime with benchmarking QLEDs, including both bottom and top emission green devices. The reference numbers are indicated within square brackets.

plays a critical role in eliminating the angular color shift. Figure 4b shows the angular distribution of the single cavity mode. Although the mode maximum slightly shifts to shorter wavelengths at large off-axis angles, the mode distribution is broad enough to cover QDs' narrow emission linewidth and ensure high integral overlap. Figure 4c shows that narrowing the QDs' FMWM from the common 27 nm to 21 nm results in a 54% reduction in the color shift ($\Delta u'v'$).

Once the emitter is selected, the value of $R_1$ should be optimized to balance the microcavity effect and total light outcoupling. Generally, a larger $R_1$ enhances the emission rate, reduces total light out-coupling, lowers the Lambertian factor, and intensifies color shifts. Figure 4d shows the dependence of EQE and CE on $R_1$. Each device was optimized for cavity resonance. As seen, although an $R_1$ of 52% generates CE values of nearly 220 cd A$^{-1}$. The maximum EQE and Lambertian factor are reduced to 22.9% and 0.48π, respectively, and the color shift at 60° increases to 0.0061. Conversely, an $R_1$ of 40% leads to a CE of only 186.6 cd A$^{-1}$, the maximum EQE and Lambertian factor are increased to 31.0% and 0.72π, respectively, and color shift at 60° reduces to 0.0046. Eventually, an $R_1 = 46\%$ was chosen for balancing CE, EQE, and color shifts. (see Supplementary Fig. 8 for the measured reflectivities of the top electrode).

With the emitter and mirrors sequentially determined, the thickness of each semiconducting layer can be finely tuned to reach the single-mode resonance. In addition to the optical effect, adjusting the charge transporting layers significantly affects the threshold voltage

and charge balance. An optical distance longer (or shorter) than 183.5 nm causes a redshifted (or blueshifted) and broadened emission spectrum (see Supplementary Fig. 9 for EL spectra due to cavity length change). Figure 4e summarizes the dependence of CE and EQE on the optical distance. Understandably, CE reaches its maximum value when the resonance condition is precisely fulfilled. A longer cavity can concentrate the emission maximum in the substrate-normal direction, leading to a higher EQE but lower CE. It is worth mentioning that these devices only differ by a few nanometers in thickness, suggesting the necessity of fine-tuning the optical parameters.

The antinode enhancement factor, $\xi$, is also critical for CE enhancement (see Eq. 1). Due to the multi-beam interference, $\xi$ reaches a maximized value of 2 when the emissive QDs are placed at the antinode of the standing wave inside the single-mode cavity. Figure 4f shows the CE of several devices resonating at the wavelength of $\lambda/2$. Shifting the emitting layer by 10 nm affects the optical gain by as much as 12%. Moreover, an ITO spacer between the bottom mirror and PEDOT:PSS is necessary to maximize the antinode enhancement factor of an electrically optimized QLED because electron-transporting oxides are optically denser than organic hole-transporting layers.

The radar chart in Fig. 4g summarizes the device's performance. Even without considering top emission's advantage in panel manufacturing, our green QLED outperforms the best BE counterpart in almost every aspect, and the color shifts of both are negligible[16,17,21]. Compared with the TE green QLEDs with complete performance data,

our device is superior in every aspect by significant margins. It is worth mentioning that most previous reports provide only CE and EQE data. Therefore, we separately compare them with our devices in Fig. 4h, i[40–45]. Figure 4h shows that our device leads in CE and EQE, even though most benchmarking devices still have lifetime and color shift issues. The lifetime comparison in Fig. 4i shows that our device surpasses all the previous BE and TE green QLEDs by simultaneously achieving superior optical and electrical performance. Our QLED demonstrates the most leading and balanced performance to date. Compared with reported planar OLEDs, our device shows significant advantages in lifetime, luminance, and color shift (see Supplementary Fig. 10).

## Discussion

We have successfully demonstrated a single-mode green QLED by combining an electrically optimum device structure with a short single-mode optical cavity. To achieve this, we resolved the issue of determining penetration depth by correcting the misused theory, and we overcame the challenge of nanocrystals' high refractive indices by developing an index-lowering process that produces uniformly distributed gaps between QDs without disrupting surface morphology. As a result, the optimized device achieves record-breaking performance metrics, including a luminance of 1.6 million nits, a current efficiency of 204.2 cd A$^{-1}$, and a $T_{95}$ lifetime of 15,600 h at 1000 cd m$^{-2}$.

We have also addressed the common concerns about the single-mode design. Lowering the mirror's reflectivity prevents excessive microcavity effects and allows sufficient light transmittance to the free space. Reducing the natural emission linewidth of the highly luminescent QDs ensures a high spectral overlap with the broader single cavity mode at different viewing angles. Consequently, unparalleled performance is achieved without compromising total light outcoupling or angular color shift. The EQE of 29.2% is the best for QLEDs without implementing light extraction methods, and the negligible color shift of $\Delta u'v' = 0.0052$ from 0 to 60° is better than previously reported multi-mode QLEDs and OLEDs.

This work demonstrates exceptional all-round QLED performance and has the potential to integrate matured light extraction methods, making it promising for both traditional and emerging applications.

## Methods

### Chemicals

Cadmium oxide (99.99%), Zinc acetate dihydrate (98%), selenium powder, sulfur powder, 1-octadecene (90%), oleic acid (90%), and ethanol (absolute, 200 Proof) were purchased from Sigma–Aldrich. Hexane (HPLC grade), trioctylphosphine (TOP, 90%), diphenylphosphine (DPP, 95%), and ethyl acetate (HPLC grade) were purchased from Aladdin. Octane was purchased from Acros Organics. The materials were used as received.

### Materials synthesis

Green-emitting quantum dots (ZnCdSe/ZnCdS/ZnS): In a 50 mL flask, 4 mmol of zinc acetate and 7 mL of oleic acid were combined and heated to 170 °C with argon flow. Then, 15 mL of 1-octadecene was added to the flask, and the temperature was raised to 300 °C. Next, a mixture of 0.5 mmol of Se (dissolved in 0.4 mL TOP and 0.05 mL DPP) with 0.15 mmol cadmium oleate (dissolved in 0.6 mL oleic acid) was injected into the flask and allowed to react for 30 min. Subsequently, 0.8 mmol of sulfur (dissolved in 0.4 mL TOP) was injected into the flask and allowed to react for 10 min. Following this, 0.3 mmol of cadmium oleate (dissolved in 1.5 mL oleic acid) was injected into the flask and allowed to react for 30 min. The solution was then cooled to 270 °C. 0.8 mmol of sulfur (dissolved in 0.4 mL TOP) was injected into the flask and reacted for 30 min. The solution was cooled to room temperature, and washed with hexane and ethanol. The precipitates were dissolved in octane for use. Some operations are critical for narrowing FHWM. First, all the precursors were pre-heated before injection, which

minimizes the temperature fluctuation. Second, the addition of DPP reduced precursor reactivity intentionally, prolonging the nucleation ripening time to 30 min, thereby facilitating Ostwald ripening.

ZnMgO nanoparticles: 2.8 mmol of zinc acetate dihydrate and 0.2 mmol of magnesium acetate tetrahydrate were dissolved in 30 mL of dimethyl sulfoxide. Subsequently, a solution of 0.6 mmol tetra-methylammonium hydroxide in ethanol was added to the precursor, followed by an hour of stirring. The solution was then centrifuged and washed with ethyl acetate. Finally, the precipitates were dispersed in ethanol for further use.

### Device fabrication

To fabricate a TE green QLED with balanced performance, the glass substrates with 10 nm thick patterned ITO underwent a sequential cleaning process in an ultrasonic bath of glass cleaner, deionized water, acetone, and isopropanol for 15 min each. They were then dried with N$_2$ before 135 nm of silver and 10 nm of ITO were sequentially deposited using magnetron sputtering. The ITO/Ag/ITO stacks were fabricated and thickness-calibrated in a TCL CSOT fab. Afterwards, the substrates underwent a 10-min UV-ozone cleaning. PEDOT:PSS was deposited using spin-coating at 7000 rpm for 40 s and then thermal annealed at 150 °C for 15 min to achieve a thickness of 20 nm. TFB (8 mg mL$^{-1}$) was spin-coated on PEDOT:PSS at 5500 rpm for 30 s and thermal annealed at 150 °C for 30 min to achieve a thickness of 22 nm. For the conventional process, the QDs (in octane, 12 mg mL$^{-1}$) were spin-coated at 3000 rpm for 40 s. The ZnMgO solution (30 mg mL$^{-1}$) was spin-coated at 6000 rpm for 30 s, and then thermal annealed at 80 °C for 30 min. For the index-lowering process, the QDs (in octane: hexane mixed solvent, 4: 1 in volume, 12 mg mL$^{-1}$) were deposited by spin-coating at 2600 rpm for 40 s to achieve a thickness of 22 nm. The ZnMgO solution (30 mg mL$^{-1}$) was spin-cast at 5200 rpm for 30 s to achieve a thickness of 36 nm, followed by film drying in a low-vacuum chamber (-1×10$^{-3}$ Pa) for 30 min. Finally, a layer of 34 nm Ag and 50 nm LiF was thermally evaporated in a high vacuum (5 × 10$^{-4}$ Pa) at a rate of 0.5 Å s$^{-1}$ and 1 Å s$^{-1}$, respectively.

For the BE device: The silver electrode requires deposition of about 100 nm and does not require evaporation of a LiF layer; The spin-coating parameters of each functional layer are slightly different to ensure that the thickness of each layer is the same as the TE device, and other operations are basically the same as TE devices. All the devices were encapsulated with cover glass and UV-curable resins (acid free) in a N$_2$-filled glovebox right after the depositions.

### Optical design

The design of a single-mode QLED was based on the resonant cavity theory. The calculation of penetration depths is detailed in Supplementary Note 1. The one-dimensional field distribution was estimated by adopting the calculated penetration depths and a Gaussian distribution function[46]. The optical mode and its angular dependence were simulated using Setfos version 4.6.5.

### Materials characterization

The thicknesses of films were measured by Atomic Force Microscope (Bruker). A spectroscopic ellipsometer (J.A. Woollam Co.) was used to obtain the raw data for ellipsometry analysis. The optical constants ($n$, $k$) were fitted using CompleteEASE (version 4.45) by considering film roughness. The transmission electron microscope images were acquired by an FEI TECNAI G20 with an acceleration voltage of 200 kV. The scanning electron microscope images were captured by a SU8010 (Hitachi). The PLQE and time-resolved PL decay transients were recorded by a FLS1000 (Edinburgh Instrument).

### Device characterization

The $L − J − V$ characteristics of the devices were measured by a programmed system comprising a source meter (Keithley 2400) for $J(V)$, a

calibrated silicon detector (Edmund) coupled with a Keithley 6485 picoammeter for photon collection, and a CCD-array spectrometer (Ocean Optics USB2000+) for emission spectra, chromaticity coordinates, and $L(V)$. Moreover, the luminance intensity in $L(V)$ and lifetime tests was independently verified using a luminance meter (Konica Minolta CS2000). The uniformity of pixel luminance intensity was confirmed to prevent the overestimation of lifetime values. For the angular dependence measurements, an optical fiber was fixed on a rotary stage with an angle scale and then connected to a USB2000+. The EQE was calculated from the luminance measured in the substrate-normal direction and the angular emission mode, and then independently verified using an integration-sphere photometer.

## Data availability

The data generated in this study are provided in the Source Data file. Source data are provided with this paper. The data that support the findings of this study are available from the corresponding author upon request.

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

## Acknowledgements

The work is supported by the National Key Research and Development Program of China (S.C., Grant 2021YFB3601700), the Pearl River Talent Recruitment Program for Guangdong Introducing Innovative and Entrepreneurial Teams (L.W., 2016ZT06C650), Priority Academic Program Development (PAPD) of Jiangsu Higher Education Institutions (S.C.), and Jiangsu Shuangchuang Plan (S.C.).

## Author contributions

M.L. contributed to the design, fabrication, and characterization of the devices. R.L. contributed to the synthesis of ZnMgO and assisted in device fabrication. L.W. and X.L. contributed to the synthesis of quantum dots. X.X. assisted in materials characterization. Z.A. contributed to optical simulation. X.S. and X.C. assisted in device characterization. S.C. and M.L. conceived the idea. S.C. supervised the experiments. The manuscript was written through the contributions of all authors. All authors have given approval to the final version of the manuscript.

## Competing interests

The authors declare no competing interests.
