## [Peer Review File · Nature Communications]

Ultrabright and Stable Top-Emitting Quantum-Dot Light-Emitting Diodes with Negligible Angular Color ShiftEditorial Note: Parts of this Peer Review File have been redacted as indicated to remove third-party material where no permission to publish could be obtained.

Reviewers' comments:

Reviewer #1 (Remarks to the Author):

The manuscript detailing high-efficiency ZnCdSe-based top-emitting green QLEDs, demonstrating an EQE of 29.2%, current efficiency of 204.2 cd/A, and a T95 operational lifetime of 15,600 h at 1000 nits, is commendable. The enhancements in device efficiency and stability, attributed to the refined manipulation of nanocrystal film reflective indices and precise optical parameter calculations, are notable. However, given the significant potential impact on subsequent research, rigorous cross-validation is necessary. The paper is suitable for publication in Nature Communications, subject to the following revisions:

The reported CE/EQE ratio (approximately 7.0) surpasses current benchmarks in green-emitting QLEDs. To substantiate these figures, independent verification using a spectroradiometer, like the CS-2000, is advised. This would enable simultaneous assessment of luminance and emission spectra, thus reinforcing the validity of the IVL measurements.

Inclusion of statistical data on device characteristics, such as EQE distribution, is essential for a comprehensive understanding.

The paper should address additional details for comprehensiveness, clarity, and reproducibility:

- a) The energy level diagram, including the CBM and VBM levels of the QD emission layer, is absent and should be included.
- b) Surface morphology of the QD emission layer, through SEM and AFM imaging, is required.
- c) Clarification on the use of an octane:hexane mixed solvent in the index-lowering process is necessary.
- d) Current density values at luminance levels of 100 and 1,000 nits in Figure 2C should be specified.
- e) Insight into the sub-threshold turn-on voltage (1.9 V) origin is required.
- f) The Materials section needs more detail, including supplier information for reproducibility.
- g) Detailed data on the top mirror's reflectivity, the refractive indices, thicknesses of non-metallic layers, and distances between the emission layer and electrodes in Figures 4d,e,f, is crucial for understanding the device structure.

Reviewer #2 (Remarks to the Author):

In this manuscript, the authors have fabricated green-emitting QLEDs that exhibit record-breaking values for luminance, current efficiency, and operational lifetimes. They attribute this enhanced performance to a series of factors, including improved methods of determining penetration depths, modulation of refractive indices, reduced emission full-width half-maximum, and lowering the top mirror's reflectivity.

While the work and performance is commendable, I do not believe it merits being published

in Nature Communications. There are multiple methods listed here to increase performance, but I do not see the big, innovative idea that would push this work to the level of Nature Communications. The literature article adma.202106276, which the authors cite in Table S2, has already shown that top-emitting quantum dots can show impressive performance in single-mode, albeit in the red.

A few other general points/questions.

1. Why is the photoluminescence quantum efficiency higher in solid film than in solution?
2. The authors should comment on why their QDs have a reduced FWHM.
3. In the abstract, the authors state the negligible angular color shift is due to the narrowed emission linewidths of quantum dots. However, in the main text, they state that this work shows the “lowest color shift ever reported for top-emitting QLEDs, even though previous work generally attempted to reduce color shift using multi-mode.” Is this just because the quantum dots in this paper have a narrower linewidth than the dots used in other papers, or is there anything else at work here?

Reviewer #3 (Remarks to the Author):

Li and colleagues have introduced a compelling concept to enhance QLED light output coupling, EQE, and color toning stability through a micro cavity configuration. While the results are intriguing, there are several issues that need addressing before further consideration:

1. Introduction (Page 2):

The introduction lacks references to recent QLED works crucial for a broader readership. Please consider adding the following for context:

- Nat. Nanotechnol. 18, 1168–1174 (2023). [<https://doi.org/10.1038/s41565-023-01441-z>]
- J. Mater. Chem. C, 2020, 8, 10676-10695. [<https://doi.org/10.1039/D0TC01349H>]
- Nanoscale Horiz., 2021, 6, 68-77. [<https://doi.org/10.1039/D0NH00556H>]
- Mater. Adv., 2022, 3, 6773-6790. [<https://doi.org/10.1039/D2MA00375A>]

2. Introduction (Page 2):

In addition to Schubert's classic work (reference 12), consider recent theoretical works such as 2022 J. Phys. Mater. 5 044009 on LEE optimization in QLED design [<https://doi.org/10.1088/2515-7639/ac9e77>].

3. Results (Page 4, Lines 17-19):

Clarify why the as-prepared QDs exhibit higher thin film PLQY than the solution state. Provide a clear rationale and relevant references to support this claim, addressing potential changes in absorption spectra, such as a red shift.

4. Results (Page 4, Lines 22-24):

Explain the selection of a 10 nm ITO layer and discuss the impact of different thicknesses. Consider the suitability of FTO and elaborate on the main function of ITO.

5. Figure 1 (Page 5):

Redraw Figure 1 with specific attention to:

- Highlighting the micro cavity area in the QLED device and labeling typical thickness and design.
- Providing a focused cross-section TEM image with a clear definition of layer thickness.

- Offering a detailed explanation of the data extraction and calculation process for Figures 1d and e.

6. Page 10:

Specify the method for extracting the spontaneous emission line with $\Delta\lambda$.

7. Page 10:

Provide an optical simulation on Page 10 supporting the spontaneous emission narrowing effect. Address whether the experimental narrowing value aligns with or differs from the optical simulation (from 27 nm to 21 nm).

8. Page 12:

Clarify the definition and calculation method for the antinode enhancement factor.

9. Page 15:

Specify the version and access date of Setfos used for the optical design.

Reviewers' comments:

Reviewer #1 (Remarks to the Author):

The manuscript detailing high-efficiency ZnCdSe-based top-emitting green QLEDs, demonstrating an EQE of 29.2%, current efficiency of 204.2 cd/A, and a T95 operational lifetime of 15,600 h at 1000 nits, is commendable. The enhancements in device efficiency and stability, attributed to the refined manipulation of nanocrystal film reflective indices and precise optical parameter calculations, are notable. However, given the significant potential impact on subsequent research, rigorous cross-validation is necessary. The paper is suitable for publication in Nature Communications, subject to the following revisions:

The reported CE/EQE ratio (approximately 7.0) surpasses current benchmarks in green-emitting QLEDs. To substantiate these figures, independent verification using a spectroradiometer, like the CS-2000, is advised. This would enable simultaneous assessment of luminance and emission spectra, thus reinforcing the validity of the IVL measurements.

Response:

We appreciate the positive comment. Regarding the high CE/EQE ratio, please see the explanations below.

1. CE is defined using the luminance intensity in the substrate-normal direction, while EQE counts all the photons emitted to the free space.

Due to the QDs' narrow emission linewidth, a *bottom-emitting* ZnCdSeS-based green QLED generally produces CE/EQE ratios of up to 4.0~4.5, depending on the emission wavelength (*Nature Photonics* 2022, 16 (7), 505-+; *Nature Photonics* 2018, 12 (3), 159-164.). A well-designed *top-emitting* green QLED can show even higher CE/EQE ratios because a moderate microcavity effect can substantially enhance the emission rate in the substrate-normal direction. Meanwhile, EQE is usually less enhanced, which causes a larger CE/EQE ratio. Our work demonstrated the first single-mode *top-emitting* green QLED, which has stronger CE enhancement than a multi-mode cavity. The CE/EQE ratio of **6.84** (EQE=29.2%, CE=204 cd A⁻¹ is a **1.6-fold enhancement** compared with its *bottom-emitting* counterpart.

2. The CE/EQE ratio of 7 is reasonable for green LEDs with narrow emission linewidths.

For example, bottom-emitting green OLEDs show a CE/EQE ratio of 1-3.5, depending on the emission spectra (*Appl. Phys. Lett.* 91, 183516 (2007)), **top-emitting green OLEDs with narrow emission bandwidth** can show CE/EQE ratios of up to **7.2** (EQE=30.5%, CE=220 cd A⁻¹. *Angew. Chem. Int. Ed.* 2022, 61, e202202380.), offering a >2-fold enhancement in the CE/EQE ratio.

3. We have been doing data crosschecks with TCL's QLED team on a regular basis.

We understand the reviewer's request for data crosscheck stems from the concern

of the CE/EQE ratio. Therefore, we believe the above explanation should address the issue.

Moreover, we would like to share that our group has been doing data crosschecks with TCL's QLED team, which has CS-2000, on a regular basis. Below is the result of a similar device that we sent to TCL in December 2023. Please note that the purpose of showing this data is to prove that the CE/EQE ratio is not due to instrumental issues.

Figure. A similar device measured by TCL on December 18, 2023. **a** EL spectra; **b** current efficiency; **c** external quantum efficiency; **d** Photos of CS-2000.

Table 1. Summary of Device Performance.

EL peak (nm)	FWHM (nm)	CIE-x	CIE-y	CE (cd A ⁻¹)	EQE (%)	T ₉₅ @1000 cd m ⁻² (h)
537	19.1	0.227	0.751	210.8	30.2	16000

Inclusion of statistical data on device characteristics, such as EQE distribution, is essential for a comprehensive understanding.

Response: We appreciate the comment. EQE distribution has been provided in the revised SI.

Figure S3b EQE distribution of 20 top-emitting devices with optimal parameters

The paper should address additional details for comprehensiveness, clarity, and reproducibility:

a) The energy level diagram, including the CBM and VBM levels of the QD emission layer, is absent and should be included.

Response:

We appreciate the comment. The energy diagram has been provided in the revised SI (Figure S3). It is worth noting that the energy levels of isolated materials play a limited guiding role in the design of the QLED devices due to the disorder-induced electrostatic phenomena (detailed mechanism can be found in our recent publications: *The Journal of Physical Chemistry Letters* 2023, 14 (20), 4830-4836; *Device* 2023, 1 (3), 100061.)

Figure S3c Schematic diagram of energy level arrangement of top-emitting devices

b) Surface morphology of the QD emission layer, through SEM and AFM imaging, is required.

Response:

We appreciate the comment. The surface morphology measured by SEM has been provided in the revised SI. The SEM image shows that the film prepared using the index-lowering process results in uniformly distributed gaps of several nanometers wide between QDs without disruptive cracks and pinholes. This leads to a 12% decrease in the packing density within the plane, which can be attributed to the suppression of nanocrystal reassembling. The reduction in packing density successfully lowers the refractive indices. To the best of our knowledge, similar methods have not been reported for solution-processed LEDs.

Figure 1d in revised MS Left: QD film prepared using the conventional process, the closely

packed QDs offers an areal density of 53.5 ± 1.6 particles per 10000 nm^2 ; Right: QD film prepared using the index-lowering process, areal density is reduced 47.8 ± 2.0 particles per 10000 nm^2 (the arrows mark a few nanometer-wide gaps, which are uniformly distributed within the plain). The ratio in areal density (53.5 vs. 47.8) matches the difference in refractive indices (1.9 vs. 1.75).

Besides, we need to clarify that measuring the surface morphology of QD films using AFM has been challenging. The non-complete organic ligand coverage may cause inconsistent interaction between the sample surface and the AFM tip. **In our case, AFM can hardly resolve the nanometer-level features on the surface of QD films.** As seen below, the same samples that provided the SEM images show much less difference under AFM. Therefore, we prefer using SEM data in the revised MS.

Figure: AFM height diagram obtained from the same samples as the SEM measurement.

c) Clarification on the use of an octane:hexane mixed solvent in the index-lowering process is necessary.

Response: The refractive index of nanocrystal-assembling films is determined by the refractive index of the corresponding bulk materials and the packing density of nanocrystals. Adding a low-boiling-point solvent (hexane) can accelerate the solvent evaporation rate, which lowers the packing density without disrupting the morphology. Additionally, replacing post-thermal treatment with low-vacuum drying can suppress the reassembling of nanocrystals. The above processes jointly lower the packing density of QDs by 12%, and lower the film's refractive index by more than 0.15. **Please refer to the SEM images we showed in the response to comment b)** for the difference in packing density.

d) Current density values at luminance levels of 100 and 1,000 nits in Figure 2C should be specified.

Response: The current density at 100 and 1000 nits are 0.2 and 1.1 mA cm^{-2} , respectively, as seen in Figure S4. Figure 2c shows the actual measured luminance decay with an initial luminance of around 30000 nits (accelerated test, constant current density), instead of 100 or 1000 nits. The lifetime at 100 and 1000 nits were calculated using a widely used equation $L_0^n \cdot T_{95} = \text{constant}$, in which the acceleration factor (n) was determined to be 1.85, as seen in Figure S4. As commonly observed for QLEDs, the luminance efficiency at or less than 100 nits is usually lower than that at higher luminance, whose effect on the device degradation is reflected by the acceleration factor.

e) Insight into the sub-threshold turn-on voltage (1.9 V) origin is required.

Response:

We appreciate the comment. Sub-threshold turn-on has been a *commonly observed* phenomenon for QLEDs since 2011. In the particular case of green QLEDs, multiple groups have reported that bottom-emitting devices show turn-on voltages ~ 1.9 V (*Nature Photonics* 2011, 5 (9), 543-548; *Nature Photonics* 2022, 16 (7), 505-+; *Nature Photonics* 2019, 13 (3), 192-197.).

In 2019, we published a paper on the origin of sub-threshold turn-on, which concluded that the luminance threshold voltage is consistent with the flat band voltage of the emission layer, suggesting no energy-up-conversion. (Origin of Subthreshold Turn-On in Quantum-Dot Light-Emitting Diodes. *ACS Nano* 2019, 13 (7), 8229-8236. Citation: 41). The viewpoint on sub-threshold turn-on has been accepted by scientists like V. Klimov. We have mentioned this mechanism in the revised MS.

f) The Materials section needs more detail, including supplier information for reproducibility.

Response: The information has been provided in the revised manuscript.

g) Detailed data on the top mirror's reflectivity, the refractive indices, thicknesses of non-metallic layers, and distances between the emission layer and electrodes in Figures 4d,e,f, is crucial for understanding the device structure.

Response:

Figure S8 provides the top mirror's reflectivity (and its thickness dependence). We also specified the top mirror's thickness in the main text. The parameters regarding Figures 4d-f are available in Table S1.

Table S1 in revised SI: Optical parameters

Material	$n+ik$ = 537 nm	Fig 4d (nm)	Fig 4e (nm)	Fig 4f (nm)
Ag (bottom)	0.05+3.32i	110	110	110
Ag (top)	0.133+3.07i	30/34/38	34	34
ITO	1.93	10	10	10
PEDOT:PSS	1.56	20	18/20/22	17/20/22
TFB	1.76	22	22	22
QDs	1.75	22	22	22
ZnMgO	1.55	36	33/36/39	39/36/34

Reviewer #2 (Remarks to the Author):

In this manuscript, the authors have fabricated green-emitting QLEDs that exhibit record-breaking values for luminance, current efficiency, and operational lifetimes. They attribute this enhanced performance to a series of factors, including improved methods of determining penetration depths, modulation of refractive indices, reduced emission full-width half-maximum, and lowering the top mirror's reflectivity.

While the work and performance is commendable, I do not believe it merits being published in Nature Communications. There are multiple methods listed here to increase performance, but I do not see the big, innovative idea that would push this work to the level of Nature Communications. The literature article adma.202106276, which the authors cite in Table S2, has already shown that top-emitting quantum dots can show impressive performance in single-mode, albeit in the red.

Response: We appreciate that the reviewer acknowledged the all-round excellent device performance. Still, we believe there has been a miscommunication regarding the goal, challenges, and method in this work.

1. Achieving single-mode green QLEDs is a significant advance over earlier work. This is because it is a goal set by the measurements of panel production and involves addressing unique challenges not encountered in earlier research.

(1) Display panel manufacturing requires top emission. Among all the top-emitting designs, the single-mode resonance offers the highest brightness and stability. **Achieving single-mode resonance for R/G/B QLEDs, respectively, is a goal set by panel production rather than an optional method to improve QLED performance.** Thus, the past demonstration of a single-mode red QLED (adma.202106276) does not change the significance of single-mode green QLED, and vice versa.

(2) **Achieving the goal of single-mode green QLEDs faces unique and substantial challenges not encountered in earlier research on red devices, which makes green QLEDs and single-mode design seemingly incompatible.** The challenge mainly comes from the difference in emission wavelength and optical cavity lengths (265 nm vs. 315 nm). For a single-mode green QLED, the optical distance of all the semiconducting layers plus the reflection penetration depths at mirrors is as short as 265 nm, meanwhile ensuring electrical performance. Past attempts to meet this condition were hindered by **theoretical confusion and technical challenges**. First, the **incorrect theoretical calculation** (Light: Sci. & Appl. 2020, 9 (1), 89. & more) made researchers believe that the reflection penetration depth is too large to achieve a single-mode green QLED (see Supplementary Note 1). Second, the **high refractive indices of nanocrystals** make it difficult to shorten the total optical distance.

(3) A topic analogous to this work is QLED operational lifetime. First, achieving a long lifetime for red, green, and blue QLEDs, respectively, is a goal set by panel production. Red QLEDs' lifetimes have been long enough for industrialization since 2017. However, green/blue QLEDs show poorer lifetimes due to unique degradation mechanisms. Therefore, the community still appreciates the recent progress in green/blue QLEDs' operational lifetime (*Nat. Photonics* 2022, 16 (7), 505-+; *Nat. Commun.* 2023, 14 (1), 284.)

2. The comment "multiple methods" indicates that the research method used in the paper was not properly communicated to the reviewer. In fact, achieving the first single-mode green QLED is attributed to a theory correction and an original index-lowering process. The rest of the "multiple methods" were introduced to address the side effects of single-mode designs.

- (1) We believe that correcting the calculation theory of penetration depths can resolve a fundamental issue that has limited the industrialization of top-emitting QLEDs, which makes the impact "big".
- (2) We believe that our index-lowering process is an "innovative" way to modulate the pecking density and refractive index of nanocrystal films. We have provided new data and rewritten the paragraph on the index-lowering process in the revised MS. The SEM images (please see below, Figure 1d in the revised MS) show that the index-lowering process results in uniformly distributed gaps of several nanometers wide between QDs without forming disruptive cracks or pinholes. The method has not been previously used for making LEDs, making it "original". The 12% decrease in in-plane packing density is consistent with ~10% reduction in refractive indices. Analogously, glancing-angle deposition (GLAD) has been widely used to prepare porous and low-index films since the last century and still generates high-impact work (*Nature Photon* 1, 176–179 (2007); *Nano Lett.* 2006, 6, 4, 854–857), although tilting the substrate did not appear "innovative & big" at first glance.
- (3) As highlighted in the revised MS, other "multiple methods" were necessary to address the concerns (side effects) of the single-mode design, which includes color shifts and EQE. Without other "multiple methods," we can achieve even higher luminance intensity, current efficiency, and operational lifetime, but we chose to demonstrate an all-round performance that can satisfy our industrial collaborators.

Figure 1d in revised MS Left: QD film prepared using the conventional process, the closely packed QDs offers an areal density of 53.5 ± 1.6 particles per 10000 nm^2 ; Right: QD film prepared using the index-lowering process, areal density is reduced 47.8 ± 2.0 particles per 10000 nm^2 (the arrows mark a few nanometer-wide gaps, which are uniformly distributed within the plain). The ratio in areal density (53.5 vs. 47.8) matches the difference in refractive indices (1.9 vs. 1.75).

A few other general points/questions.

1. Why is the photoluminescence quantum efficiency higher in solid film than in solution?

Response:

As detailed in Figure S1, the QD films deposited on the specially designed oxide/metal stacks exhibit *external* PLQE from 7% to 32%. Switching to a glass substrate offers external PLQEs of 67%, which could vary due to the substrate's optical constants, thickness, and instrument. **These values are lower than those of the solution sample (86.2%).**

Thin film samples experience significant optical loss, including re-absorption of QDs, the loss due to the back-reflecting mirrors, and waveguide loss (Schnitzer et al. *Appl. Phys. Lett.* 62, 131-133 9 (1993)). To extract the *internal* PLQE, we used a method proposed by Schnitzer et al. This method has also been used in recent work for claiming record-high *internal* PLQE of luminescent thin films. **Some of them also show higher *internal* PLQE than the *external* one measured from solution samples.** (Jagielski et al. *Sci. Adv.* 2017, 3, eaaq0208; Abdi-Jalebi, M. et al. *Nature* 2018, 555, 497–501; Braly, I. L. et al. *Nat. Photonics* 2018, 12, 355–361.)

Therefore, the *internal* PLQE of 90.3% was fitted from multiple measured *external* PLQE data points using a commonly accepted method that excludes the optical losses in thin film characterization. Despite many groups having used this method to claim record-high PLQE of luminescent thin films, we still believe one should be cautious about the calculated value because the method assumes constant optical constants. In comparison, the *external* PLQE of the solution sample (86.2%) did not subtract the minimal loss. We showed the thin film's PLQE only to demonstrate that the solid-state film is highly luminescent, rather than trying to claim any record or suggest aggregation-induced emission.

2. The authors should comment on why their QDs have a reduced FWHM.

Response:

We assume that the reviewer asked why our green QDs showed a much reduced FWHM (21 nm) than previous reports (>26 nm). Green QDs (ZnCdSeS) that offered competitive QLED performance generally show FWHMs larger than 26 nm (**26 nm:** *Nat. Photonics* 2022, 16 (7), 505-+; **47 nm:** *Nat. Photonics* 2019, 13 (3), 192-197.; **29 nm:** *Nat. Photon.* 2015, 9 (4), 259-266.). Compared to these benchmarks, we achieved significantly reduced FWHM. **It is established that the FWHM of QDs is more affected by inhomogeneous broadening—distribution of nanocrystal size. Our QDs have a more even distribution of nanocrystal size because:**

- (1) We minimized the temperature fluctuation during synthesis, which includes pre-heating the precursor solution before injecting it into the reaction vessel.
 - (2) By mixing in DPP, we could prolong the nucleation ripening time (from 10 to 30 min), which converges the particle sizes of the quantum dots through the Ostwald Ripening.
 - (3) We prolonged the shell growth time to 40 min.
- (1)~(3) are *novel* compared to previous publications on green QDs, and they have been provided in the revised MS (please see method section).**

3. In the abstract, the authors state the negligible angular color shift is due to the

narrowed emission linewidths of quantum dots. However, in the main text, they state that this work shows the “lowest color shift ever reported for top-emitting QLEDs, even though previous work generally attempted to reduce color shift using multi-mode.” Is this just because the quantum dots in this paper have a narrower linewidth than the dots used in other papers, or is there anything else at work here?

Response:

We appreciate the comment.

Narrow FWHM suppresses the emission rate enhancement at other wavelengths, which leads to smaller color shifts. Ideally, zero color shift is expected if one has a monochromatic emitter. The MS has been modified as follows:

- (1) The content quoted by the reviewer might be misleading. It has been modified to *"To our best knowledge, this is the lowest color shift ever reported for top-emitting QLEDs, even though previous work generally employed multi-mode cavities and scattering layers that are beneficial for reducing color shifts."*
- (2) In theory, the single-mode cavity has a stronger microcavity effect, which provides a larger emission enhancement (benefit) and a larger color shift (side effect). In this work, **we not only addressed this concern/side effect but also achieved even lower angular color shifts than multi-mode by reducing the FWHM of QDs.** As mentioned in the revised MS, an FWHM of 21 nm is much narrower than typically reported values. Green QDs (ZnCdSeS) that offer competitive QLED performance generally show FWHMs larger than 26 nm (**26 nm**: *Nat. Photonics* 2022, 16 (7), 505-+; **47 nm**: *Nat. Photonics* 2019, 13 (3), 192-197.; **29 nm**: *Nat. Photonics* 2015, 9 (4), 259-266.).
- (3) **We have also enclosed a theoretical calculation in the revised MS to support the experimental finding** (see the caption of Figure S7). The calculation was done by multiplying QDs' emission spectrum $I(\lambda)$ with the angular-dependent optical mode density $\rho(\lambda)$ (such as Figure 4b). The resultant spectrum, if normalized, can be further converted to a CIE-1976 coordinate (u',v') for the evaluation of color shifts. As seen in the Table below, the calculated result is consistent with experiments, validating the effect of narrowing FWHM.

QDs	angular color shift $\Delta(u',v')$	
	Calculation	Experiment
FWHM=27 nm	0.0039 off angle= 30° 0.0086 off angle= 60°	0.0036 off angle=30° 0.0080 off angle= 60°
FWHM=21 nm	0.0032 at 30° 0.0060 off angle= 60°	0.0030 at 30° 0.0052 off angle= 60°

Reviewer #3 (Remarks to the Author):

Li and colleagues have introduced a compelling concept to enhance QLED light output coupling, EQE, and color toning stability through a micro cavity configuration. While the results are intriguing, there are several issues that need addressing before further consideration:

1. Introduction (Page 2):

The introduction lacks references to recent QLED works crucial for a broader readership. Please consider adding the following for context:

- Nat. Nanotechnol. 18, 1168–1174 (2023). [<https://doi.org/10.1038/s41565-023-01441-z>]

- J. Mater. Chem. C, 2020, 8, 10676-10695. [<https://doi.org/10.1039/D0TC01349H>]

- Nanoscale Horiz., 2021, 6, 68-77. [<https://doi.org/10.1039/D0NH00556H>]

- Mater. Adv., 2022, 3, 6773-6790. [<https://doi.org/10.1039/D2MA00375A>]

Response: we would like to include some of the suggested references.

The first reference is about the thermal management of red QLED, which could be considered as progress that attracts a broader readership.

2. Introduction (Page 2):

In addition to Schubert's classic work (reference 12), consider recent theoretical works such as 2022 J. Phys. Mater. 5 044009 on LEE optimization in QLED design [<https://doi.org/10.1088/2515-7639/ac9e77>].

Response:

We have cited the suggested reference as it discussed light extraction in QLED.

3. Results (Page 4, Lines 17-19):

Clarify why the as-prepared QDs exhibit higher thin film PLQY than the solution state. Provide a clear rationale and relevant references to support this claim, addressing potential changes in absorption spectra, such as a red shift.

Response:

As detailed in Figure S1, the QD films deposited on the specially designed oxide/metal stacks exhibit *external* PLQE from 7% to 32%. Switching to a glass substrate offers *external PLQEs of 67%*, which could vary due to the substrate's optical constants, thickness, and instrument. These values are lower than those of the solution sample.

Thin film samples experience significant optical loss, including re-absorption of QDs, the loss due to the back-reflecting mirrors, and waveguide loss (Schnitzer et al. *Appl. Phys. Lett.* 62, 131-133 9 (1993)). To extract the *internal* PLQE, we used a method proposed by Schnitzer et al. This method has also been used in recent work for claiming record-high *internal* PLQE of luminescent thin films. *Some of them also show higher internal PLQE than the external one measured from solution samples.* (Jagielski et al. *Sci. Adv.* 2017, 3, eaaq0208; Abdi-Jalebi, M. et al. *Nature* 2018, 555, 497–501; Braly, I. L. et al. *Nat. Photonics* 2018, 12, 355–361.)

Therefore, the *internal* PLQE of 90.3% was fitted from multiple measured *external* PLQE data points using a commonly accepted method that excludes the optical losses in thin film characterization. Despite many groups having used this method to claim record-high PLQE of luminescent thin films, we still believe one should be cautious about the calculated value because the method assumes constant optical constants. In comparison, the *external* PLQE of the solution sample (86.2%) did not subtract the minimal loss. We showed the thin film's PLQE only to demonstrate that the solid-state film is highly luminescent, rather than trying to claim any record or suggest aggregation-induced emission.

Regarding the comment on redshift, we guess the reviewer was indicating that the thin film samples possibly show a redshift in the absorption edge compared to that of the solution samples. However, that possible redshift was not observed in our samples. This is also common for colloidal QDs.

4. Results (Page 4, Lines 22-24):

Explain the selection of a 10 nm ITO layer and discuss the impact of different thicknesses. Consider the suitability of FTO and elaborate on the main function of ITO.
Response:

The 10 nm ITO was deposited on top of Ag for two reasons: (1) PEDOT: PSS solution (in DI water) shows **poor wettability** on the Ag substrate, deteriorating the film quality; (2) Spin-coating PEDOT:PSS (pH value <3) causes **a possible chemical reaction with Ag**, which changes the optical property of the interface.

We need the ITO layer to shield Ag from the acid PEDOT: PSS solution and improve the wettability. However, the ITO layer is also part of the microcavity. Therefore, **excessive ITO thickness consumes the optical length of the cavity**, making it even less probable for a green QLED to achieve single-mode resonance. Therefore, we picked a minimum (10 nm) to protect Ag, improve wettability, and, more importantly, minimize the consumption of optical distance.

5. Figure 1 (Page 5):

Redraw Figure 1 with specific attention to:

- Highlighting the micro cavity area in the QLED device and labeling typical thickness and design.

Response:

The top and bottom mirrors and semiconducting layers form a microcavity. Each layer's thickness, measured by AFM, has been provided in Figure S3d. Each thickness was averaged from three measurements. The box widths denote the data range.

- Providing a focused cross-section TEM image with a clear definition of layer thickness.

Response:

We appreciate the comment. In the revised MS, we have provided a cross-sectional TEM image to replace the original cross-section SEM image in Figure 1c.

As commonly accepted, the interface between organic layers (PEDOT:PSS/TFB) cannot be clearly resolved. Therefore, we have additionally provided the thickness calibration data (AFM, step profile, see Figure S3d, also see below) after depositing each functional layer. The ITO/Ag/ITO stacks below the PEDOT:PSS layer were fabricated and thickness calibrated in a TCL CSOT fab.

- Offering a detailed explanation of the data extraction and calculation process for Figures 1d and e.

Response:

The data extraction method has been provided in the revised MS (method section).

In Figure 1d, the sketch of the mode distribution was determined using the calculated field penetration depths (Supplementary Note 1) and the assumption that a single mode presents a Gaussian distribution inside the cavity (Appl. Phys. Lett. 93, 031110 (2008)). The sketch is used to demonstrate the location of mode maximum,

rather than provide a strict numerical solution.

In Figure 1e, the n, k values were fitted from the raw data of ellipsometry (M2000 (J.A. Woollam Co.)) using the software CompleteEASE (version 4.45). The ellipsometer and the data processing software are widely accessible.

6. Page 10:

Specify the method for extracting the spontaneous emission line with $\Delta\lambda$.

Response:

In common practice, colloidal QDs' emission linewidth can be evaluated using FWHM, which was measured using a spectrometer in our work. Therefore, a reduction in FWHM (PL and EL) corresponds to a reduction in $\Delta\lambda$ as long as the QD sizes are mono-dispersed.

We understand that the definition of linewidth may differ with the change of peak profile. Therefore, throughout the manuscript, we did not calculate any $\Delta\lambda$ value or use this value to calculate the emission rate enhancement using Equation 1. Equation 1 was enclosed to introduce the microcavity effect.

7. Page 10:

Provide an optical simulation on Page 10 supporting the spontaneous emission narrowing effect. Address whether the experimental narrowing value aligns with or differs from the optical simulation (from 27 nm to 21 nm).

Response: The term "spontaneous emission narrowing effect" may apply to two different effects. Here, we explain both in case of miscommunication.

First, we show that the emission spectrum narrowing due to the microcavity effect is consistent between theory and experiment.

Colloidal QDs	simulated QLED FWHM	measured QLED FWHM
FWHM=27 nm	22.8 nm	22.3 nm
FWHM=21 nm	19.4 nm	19.2 nm

In addition, since the reviewer mentioned "optical simulation" and "from 27 nm to 21 nm", we assume the reviewer also meant the enhancement in CE and reduction in angular color shift caused by narrowing the QDs' FWHM from 27 nm to 21 nm.

The enhancement in CE: According to Equation 1 in MS, narrowing QDs' FWHM from 27 to 21 nm enhances the luminance in the substrate-normal direction by 20%, which is roughly consistent with the results we measured from the devices (15%).

The reduction in angular color shift: the comparison can be easily done by multiplying QDs' emission spectrum $I(\lambda)$ with the angular-dependent optical mode density $\rho(\lambda)$ (already obtained as Figure 4b). The resultant spectrum, if normalized, can be further converted to a CIE-1976 coordinate (u',v') for the evaluation of color shifts. The experiment vs theory results are summarized below. As seen, the experimental results agree well with the simulation.

QDs	CE		angular color shift $\Delta(u',v')$ (30°/60°)	
	Calculation	Experiment	Simulation	Experiment
FWHM=27 nm	1.0 (normalized)	1.0 (normalized)	0.0039/0.0086	0.0036/0.0080
FWHM=21 nm	1.2	1.15	0.0032/0.0060	0.0030/0.0052

8. Page 12:

Clarify the definition and calculation method for the antinode enhancement factor.

Response:

The reviewer might not have noticed that ξ was defined as a parameter of Equation 1, which is in the introduction section of MS.

In our work, we did not calculate ξ value. Instead, we tried to explain the effect of emitting layer location (Figure 4f) by referring to the ξ effect, which was detailed in Schubert, E. (2007). Light Emitting Diodes 2nd Edition. "*When we consider the standing wave effect in a resonance cavity LED, that is, the distribution of the optically active material relative to the nodes and antinodes of the optical wave. The antinode enhancement factor ξ has a value of 2 in Equation 1, if the active region is located exactly at an antinode of the standing wave inside the cavity. The value of ξ is unity if the active region is smeared out over many periods of the standing optical wave. Finally, $\xi = 0$ if the active material is located at a node.*"

The manuscript has been modified accordingly.

9. Page 15:

Specify the version and access date of Setfos used for the optical design.

Response: Setfos ver 4.6.5, date: 2018-05-17. We are not sure about the meaning of access date; please advise.

[Screen shot of the software redacted]

REVIEWERS' COMMENTS

Reviewer #1 (Remarks to the Author):

The revised manuscript demonstrates significant enhancements in the interpretation of experimental results with details, indicating its substantial academic and technological impact. I recommend the publication of this manuscript in Nature Communications without further review.

Reviewer #2 (Remarks to the Author):

The edits and rebuttal of the authors have mitigated my initial reservations with this work. The big idea of the "index lowering process" comes across more prominently in the revised manuscript and is tied nicely with their improved performance. They authors have also modified their text to showcase how this manuscript is specifically tackling the challenges of top-emitting green QLEDs. The performance was always impressive itself. I would recommend publishing of the current manuscript.

Reviewer #3 (Remarks to the Author):

The authors have addressed all of my questions and I believe the manuscript is ready to be accepted.